# Vitamin D Is Associated with Lipid Metabolism: A Sex- and Age-Dependent Analysis of a Large Outpatient Cohort

**DOI:** 10.3390/nu16223936

**Published:** 2024-11-18

**Authors:** Xitong Li, Yvonne Liu, Jingyun Wang, Xin Chen, Christoph Reichetzeder, Saban Elitok, Bernhard K. Krämer, Cornelia Doebis, Katrin Huesker, Volker von Baehr, Berthold Hocher

**Affiliations:** 1Fifth Department of Medicine (Nephrology/Endocrinology/Rheumatology/Pneumology), University Medical Center Mannheim, University of Heidelberg, 69120 Mannheim, Germany; lixitong0729@gmail.com (X.L.); yvonne.liuye.2000@gmail.com (Y.L.); wjy6263@126.com (J.W.); xin.chen@charite.de (X.C.); bernhard.kraemer@umm.de (B.K.K.); 2Department of Nephrology, Charité Universitätsmedizin Berlin, 10117 Berlin, Germany; 3Institute for Clinical Research and Systems Medicine, Health and Medical University, 14467 Potsdam, Germany; christoph.reichetzeder@hmu-potsdam.de (C.R.); saban.elitok@klinikumevb.de (S.E.); 4Department of Nephrology and Endocrinology, Klinikum Ernst von Bergmann, 14467 Potsdam, Germany; 5Institute of Medical Diagnostics (IMD), 12247 Berlin, Germany; c.doebis@imd-berlin.de (C.D.); k.huesker@imd-berlin.de (K.H.); v.vonbaehr@imd-berlin.de (V.v.B.); 6Reproductive and Genetic Hospital of CITIC-Xiangya, Changsha 410008, China; 7School of Medicine, Central South University, Changsha 410078, China

**Keywords:** 25-hydroxyvitamin D [25(OH)D], low-density lipoprotein (LDL), high-density lipoprotein (HDL), total cholesterol (TC), sex differences, age-dependent effects, cross-sectional study

## Abstract

**Background:** Vitamin D is a fat-soluble steroid that influences cardiovascular health by affecting lipid metabolism. Since dyslipidemia is a key risk factor for cardiovascular disease (CVD), our study aimed to explore the relationship between vitamin D levels and lipid parameters, considering the effects of age and gender. **Methods**: In this cross-sectional study of 47,778 outpatients, we analyzed correlations between two forms of vitamin D—25-hydroxyvitamin D (25(OH)D) and 1,25-dihydroxyvitamin D (1,25(OH)_2_D)—and lipid parameters, including low-density lipoprotein (LDL), high-density lipoprotein (HDL), and total cholesterol (TC). Subgroup analyses by age and gender provided additional insights. **Results**: Results showed that 25(OH)D levels were negatively correlated with LDL and TC across the cohort. This association was particularly evident in men over 50, whereas women showed a positive correlation with LDL and TC before age 50 and a negative correlation after. HDL levels positively correlated with 25(OH)D across all age groups, with the strongest association in postmenopausal women. In contrast, 1,25(OH)_2_D showed a positive correlation only with HDL in individuals over 50, with no significant correlation with LDL or TC in any age group. **Conclusions**: In conclusion, findings from this cross-sectional study underscore an association between elevated levels of 25(OH)D and more favorable lipid profiles, characterized by reduced LDL and total cholesterol, as well as increased HDL levels. This association is particularly pronounced among individuals over 50 years of age and postmenopausal women.

## 1. Introduction

Vitamin D is a fat-soluble steroid that plays a crucial role in calcium homeostasis and bone metabolism [1,2]. It exists in various forms, with total 25-hydroxyvitamin D (25(OH)D) serving as the primary storage form and 1,25-dihydroxyvitamin D (1,25(OH)_2_D) acting as the biologically active form [3]. Additionally, a small fraction exists as free 25-hydroxyvitamin vitamin D, which is not bound to proteins and is directly involved in cellular functions. These metabolites also exert distinct functional and physiological effects [4,5].

There is growing evidence that moderately elevated vitamin D levels might reduce the risk of cardiovascular disease. Considering that lipid metabolism is a key independent risk factor for such conditions, it is essential to understand the interplay between vitamin D and lipid profiles. Critical components of lipid metabolism include low-density lipoprotein (LDL), high-density lipoprotein (HDL), and total cholesterol (TC). LDL, commonly known as ‘bad’ cholesterol, serves as the primary transporter of cholesterol, moving it from the liver to peripheral tissues. Elevated levels of LDL lead to cholesterol deposition within arterial walls, thereby accelerating the development of atherosclerosis [6]. A large cross-sectional study involving over 29,000 participants from diverse countries and ethnicities identified LDL as an independent risk factor for myocardial infarction [7]. Additionally, research has demonstrated that medications aimed at lowering LDL concentrations effectively reduce the incidence of myocardial infarction and other cardiovascular diseases [8].

In contrast, HDL, commonly referred to as ‘good’ cholesterol, plays a crucial role in transporting cholesterol from peripheral tissues and arterial walls back to the liver. This process helps mitigate cholesterol accumulation in the bloodstream, thereby lowering the risk of atherosclerosis [9]. Furthermore, relatively high levels of HDL are associated with reduced cholesterol oxidation and diminished inflammation within blood vessels, positioning HDL as a key protector of cardiovascular health. TC, on the other hand, represents the complete amount of all cholesterol types in the blood, encompassing LDL, HDL, and other lipid components.

Vitamin D and cholesterol, both of which originate from the common precursor 7-dehydrocholesterol [10], are closely interconnected. It has been proposed that vitamin D may influence cholesterol synthesis by modulating the activity of HMG-CoA reductase, a critical enzyme in cholesterol biosynthesis, which converts HMG-CoA to mevalonate. By influencing this enzyme’s activity, vitamin D might indirectly regulate cholesterol synthesis and thereby impact lipid metabolism, specifically LDL cholesterol levels [11]. Additionally, vitamin D may impact HDL cholesterol levels by affecting the expression and function of the cholesterol transporter protein ABCA1, which acts as an efflux pump for the removal of lipids [12].

Recent studies have explored the relationship between vitamin D and lipid metabolism, but findings remain inconclusive. Some research suggests that vitamin D significantly influences lipid metabolism, with high vitamin D levels correlating with reduced LDL and TC concentrations [13,14] and increased HDL levels [15]. However, this association has not been consistently observed across all studies [16]. This discrepancy may arise from variations in sample sizes across different studies, as well as differences observed in subgroup analyses. Moreover, there is a lack of research addressing the impact of gender and age on the relationship between vitamin D and lipid metabolism. This paper aims to analyze extensive data from the general population in Germany to elucidate the relationship between vitamin D and lipid metabolism and to comprehensively examine how gender and age affect this relationship.

## 2. Materials and Methods

### 2.1. Study Population

This study was conducted according to the guidelines of the Declaration of Helsinki and was approved on 4 October 2023 by the Institutional Review Board of the Institute of Medical Diagnostics Berlin–Potsdam. Patient consent was waived because the data used for this study were anonymized and came from internal quality checks concerning the stability of the assays, which are carried out regularly to ensure optimal patient care. When evaluating the data, the anonymity of the subject data was fully guaranteed.

The blood samples all came from referring physicians from outpatient practices or medical care centers from all over Germany sending blood from outpatients for clinical analysis to the Institute for Medical Diagnostics (IMD) Berlin–Potsdam. All samples from March 2014 to July 2020 sent to the IMD were screened and included when the following inclusion criterion were fulfilled:Measurement of vitamin D or measurement of 1,25 (OH)_2_ vitamin D.Having data on lipid parameters such as HDL, LDL, and total cholesterol, which were analyzed in a time window of 4 weeks before or after the blood sample was taken for vitamin D determination.Having additional clinical laboratory parameters on mineral bone metabolism such as calcium, phosphate, and iPTH, which were analyzed in a time window of 4 weeks before or after the blood sample was taken for vitamin D determination.Having additional clinical laboratory data on hemoglobin, white blood cell count, and CrP.

Patients were only enrolled if vitamin D measurements (criteria 1) and lipid data (criteria 2) and at least one of criteria 3 and 4 were fulfilled.

This protocol resulted in datasets of 47,778 outpatients whose blood was analyzed at the IMD Berlin–Potsdam.

### 2.2. Clinical and Laboratory Parameters

The levels of 25(OH)D were determined using the Abbott Architect i2000 electrochemiluminescence immunoassay (Abbott Laboratories, Wiesbaden, Germany). Similarly, concentrations of 1,25(OH)_2_D were measured using the Abbott Architect i2000 Chemiluminescent Immunoassay Analyzer (Abbott Laboratories, Wiesbaden, Germany). Other parameters, including LDL, HDL, and TC, were quantified using standardized methods at the Institute for Medical Diagnostics, Berlin (https://www.imd-berlin.de/ accessed on 29 May 2024). Rigorous quality assurance and control procedures were applied to all clinical and laboratory data.

### 2.3. Statistical Analyses

Statistical analyses were performed using SPSS version 23.0 (IBM, Armonk, NY, USA). All variables were expressed as mean ± standard deviation (SD), except for gender. We conducted a seasonal analysis of the average vitamin D concentrations across each month of the year. Based on the results, the months were grouped into two categories: one with low average vitamin D concentrations (January to April and December) and the other with high average concentrations (May to November), as described in our recent study [17]. Parameters were categorized into deciles based on serum concentrations of 25(OH)D or 1,25(OH)_2_D, with mean values depicted graphically through curve estimation analyses using linear, exponential, and quadratic models. Multiple linear regression analyses were performed with 25(OH)D and 1,25(OH)_2_D as dependent variables, incorporating age, sex, seasonal grouping, LDL, and HDL as predictors. If a parameter in the model appeared to potentially have a non-linear relationship with vitamin D concentration, the square root of that parameter was included in the model to capture both linear and non-linear relationships [18]. When a positive result was observed, we employed the Benjamini–Hochberg (BH) procedure to calculate the false discovery rate (FDR) threshold, minimizing the risk of Type I errors (false positives). In this method, all *p*-values from the multiple regression analysis are first arranged in ascending order. Here, m represents the total number of *p*-values, and i denotes the index of each *p*-value corresponding to a given parameter. The FDR threshold is then determined using the formula: “FDR threshold = (i/m) × α”, where α is the pre-specified FDR control level, set at 0.05. A parameter’s independent correlation is considered statistically significant if its raw *p*-value is below the corresponding FDR threshold.

The cohort was further stratified by gender, after which the multiple regression analysis was repeated. Additionally, the male and female groups were subdivided by age into those younger than 50 years and those aged 50 years or older, followed by correlation analyses with the two vitamin D forms. Correlation graphs were generated using GraphPad Prism version 8 (GraphPad Software Inc., La Jolla, CA, USA), with values expressed as mean ± standard error of the mean (SEM). The significance threshold was set at *p* < 0.05.

## 3. Results

The study cohort comprised 47,778 participants, of which 53.6% (25,617) were women. The mean age of the cohort was 59.00 years. Average levels of the key parameters were as follows: 25(OH)D at 29.60 ng/mL, 1,25(OH)_2_D at 51.60 pg/mL, LDL at 116.42 mg/dL, HDL at 55.23 mg/dL, and TC at 187.99 mg/dL. Additional measured parameters included serum calcium at 2.35 mmol/L, serum phosphorus at 1.09 mmol/L, intact parathyroid hormone (iPTH) at 58.16 pg/mL, and creatinine at 1.18 mg/dL. Detailed characteristics of the cohort are represented in Table 1, and the distribution of key parameters is illustrated in Figure 1.

**Table 1 nutrients-16-03936-t001:** Baseline characteristics of the study population.

Parameters	Reference Range	N	Mean ± SD
Gender	-	47,778 Males: 22,161 (46.4%) Females: 25,617 (53.6%)	-
Age (years)	-	47,778	59.00 ± 19.51
25(OH)D (ng/mL)	30.00–100.00	47,778	29.60 ± 13.13
1,25(OH)_2_D (pg/mL)	19.90–79.30	1927	51.60 ± 25.62
LDL (mg/dL)	<115.00	32,074	116.42 ± 40.20
HDL (mg/dL)	>45.00	31,311	55.23 ± 17.73
Total Cholesterol (mg/dL)	<200.00	30,844	187.99 ± 45.52
Calcium (mmol/L)	1.90–2.75	47,778	2.35 ± 0.11
Phosphate (mmol/L)	0.81–2.42	47,778	1.09 ± 0.22
iPTH (pg/mL)	15.00–65.00	28,116	58.16 ± 32.77
Creatinine (mg/dL)	0.16–1.95	40,496	1.18 ± 0.62
Hemoglobin (g/dL)	12.00–17.00	38,889	13.47 ± 1.57
White blood cells (Gpt/L)	3.60–28.20	38,896	7.04 ± 2.33
C-reactive protein (mg/L)	<5.00	35,681	4.25 ± 1.07

The reference range is from the Berlin Institute for Medical Diagnostics (https://www.imd-berlin.de, accessed on 29 May 2024). Abbreviations: 25(OH)D: 25-hydroxy-vitamin D; 1,25(OH)_2_D: 1,25-dihydroxy-vitamin D; LDL: low-density lipoprotein, HDL: high-density lipoprotein.

**Figure 1 nutrients-16-03936-f001:**
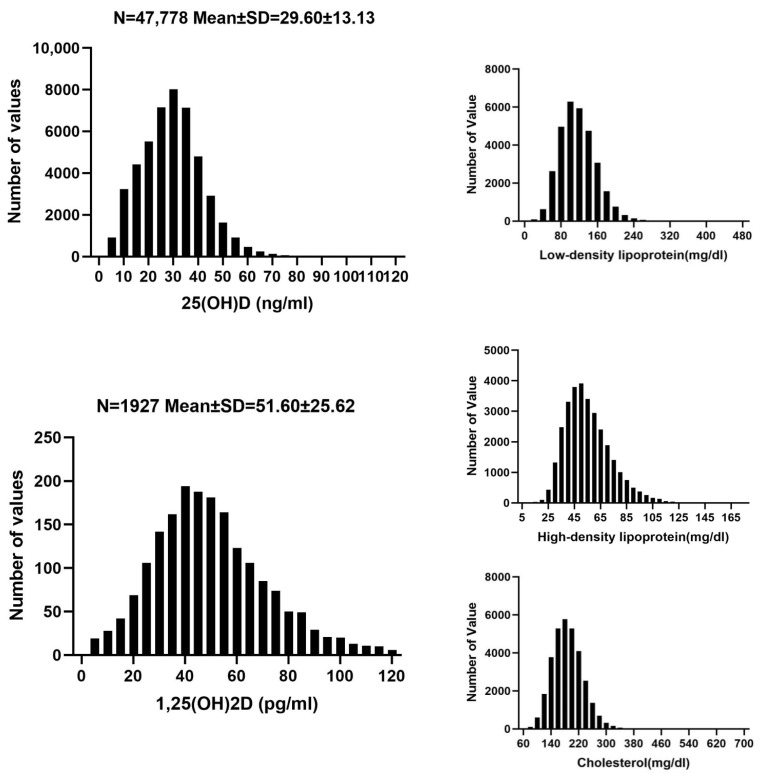
Distribution of all parameters. Abbreviations: 25(OH)D: 25-hydroxy-vitamin D; 1,25(OH)_2_D: 1,25-dihydroxy-vitamin D. Various curve-fitting models, including linear, exponential, and quadratic models, were employed to examine the relationship between vitamin D levels and lipid profiles. For 25(OH)D, both linear and quadratic models demonstrated high R^2^ values, indicating a predominantly linear relationship with LDL, HDL, and TC. In contrast, only HDL showed a notable correlation with 1,25(OH)_2_D, with the quadratic model providing the best fit, as detailed in Table 2 and Table 3.

Further analyses, with 25(OH)D and 1,25(OH)_2_D concentrations assessed in 10% increments and grouped into ten categories, showed that 25(OH)D had significant negative correlations with LDL and TC and a positive trend with HDL. In contrast, 1,25(OH)_2_D demonstrated a significant positive trend with HDL but showed no significant correlation with LDL or TC (Figure 2). These trends remained consistent across both male and female subgroups, as shown in Figure 3. Interestingly, at comparable vitamin D levels, lipid parameters were consistently higher in women than in men.

We then conducted multiple regression analyses separately for 25(OH)D and 1,25(OH)_2_D. For 25(OH)D, combining the raw *p*-values and their comparison with the corresponding FDR thresholds, the results indicated a strong independent correlation with both LDL and HDL, a trend that remained consistent even when stratifying the cohort by gender (Table 4). In contrast, for 1,25(OH)_2_D, we incorporated the square root of LDL and HDL into the model to account for potential non-linear relationships. The analysis showed no independent correlation between 1,25(OH)_2_D and either LDL or HDL, both in the overall cohort and when stratified by gender, as presented in Table 5.

We also analyzed the age-dependency by dividing the subjects into two age groups for comparative analysis. For 25(OH)D, a positive correlation with HDL was evident across all age groups for both men and women. The relationships of LDL and TC with 25(OH)D, however, were more complex. In men, no significant association with 25(OH)D was observed until age 50, with levels remaining stable as 25(OH)D increased. In men over 50, 25(OH)D was inversely associated with LDL and TC. Interestingly, in women, 25(OH)D correlated positively with LDL and TC below age 50, with this trend reversing to a negative correlation after age 50. See Figure 4 and Appendix A for details.

For 1,25(OH)_2_D, a positive correlation with HDL was observed only among individuals aged 50 and older, indicating a potential age-related effect in this association. In contrast, no significant correlation with HDL was found in the younger group under 50 years. Furthermore, across all age groups and sexes, no correlation was detected between 1,25(OH)_2_D and either LDL or TC, suggesting that 1,25(OH)_2_D levels may have limited association with these lipid parameters in the general cohort (Figure 5 and Appendix A).

To provide a comprehensive overview, a summary figure summarizing all the results is presented at the end of this article. This figure highlights the key relationships and conclusions derived from the findings (Figure 6).

## 4. Discussion

This study demonstrates a significant relationship between vitamin D levels and lipid metabolism, highlighting the interplay between 25(OH)D and lipid parameters such as LDL, HDL, and TC. Specifically, 25(OH)D was inversely correlated with LDL and TC, while showing a positive correlation with HDL. Interestingly, these relationships varied with gender and age: women displayed higher lipid levels at comparable vitamin D concentrations, and age influenced the degree of correlations. This study underscores the potential role of vitamin D in modulating lipid profiles, suggesting that assessing and, if needed, optimizing vitamin D status could be beneficial in managing lipid-related cardiovascular risk. These findings add to the growing body of evidence suggesting that vitamin D is not only crucial for bone health but also plays a significant role in cardiovascular health by influencing lipid metabolism. This research provides a comprehensive analysis of the interaction between vitamin D and lipids, emphasizing the need to consider gender and age when evaluating these effects.

Our study complements the findings of De Matteis et al. by exploring the relationship between vitamin D levels and lipid profiles, whereas De Matteis et al. focused on the association between vitamin D and monocyte-to-HDL ratio (MHR) as an inflammatory marker [19]. Both studies highlight vitamin D’s potential role in metabolic health, with our research suggesting an impact on lipid metabolism and De Matteis et al. identifying MHR as a predictor of vitamin D deficiency in various metabolic states. Unlike that of De Matteis et al., however, our study lacks direct measures of systemic inflammation (such as MHR), which limits a complete analysis of vitamin D’s anti-inflammatory effects within our cohort. Together, these findings support further investigation into vitamin D’s role in both lipid metabolism and inflammation across diverse metabolic conditions.

Our study demonstrated that overall, serum levels of both LDL and TC decreased with increasing concentrations of 25(OH)D in both men and women, a finding consistent with prior studies [20,21]. However, we did not observe a similar trend for 1,25(OH)_2_D. Hydroxylated derivatives of 25(OH)D effectively inhibit HMG-CoA reductase activity, which catalyzes the breakdown of 7-dehydrocholesterol to produce LDL and TC, thereby influencing the production of these lipid metabolites [22]. Interestingly, this inhibitory effect on HMG-CoA activity was not observed with 1,25(OH)_2_D [23]. In addition, we identified a significant positive correlation between HDL levels and both 25(OH)D and 1,25(OH)_2_D, regardless of gender and age differences. This may be due to their roles as the two primary forms of vitamin D in the body, both of which regulate the expression of the ATP-binding cassette transporter A1 gene. This gene facilitates the transport of cholesterol from cells, such as macrophages, to HDL particles, thereby enhancing both HDL levels and functionality [24].

The current study also found that women exhibited significantly higher lipid levels than men at the same vitamin D concentrations. This discrepancy can be attributed to intrinsic factors such as fat distribution [25] and metabolic differences [26,27], which generally result in higher lipid levels in women compared to men. Additionally, estrogen plays a crucial role in lipid metabolism by inhibiting HMG-CoA reductase activity, which effectively reduces lipid production and offers cardiovascular protection [28]. However, given that the average age of the women in this study was 59.45 years, the age-related decline in estrogen levels leads to reduced control over lipid metabolism, contributing to a significant increase in lipid levels in women.

We also observed a significant influence of age on the relationship between 25(OH)D and both LDL and TC levels. On a molecular level, sterol regulatory element-binding protein (SREBP) plays a critical role in lipid metabolism. SREBP forms a complex with SCAP at the membrane of the endoplasmic reticulum (ER). In conditions where cellular cholesterol levels are low, SCAP functions as a cholesterol sensor, facilitating the transport of SREBP to the Golgi apparatus. There, SREBP is cleaved and activated, allowing it to translocate to the nucleus and initiate the transcription of genes responsible for lipid and cholesterol synthesis [29]. Recent studies suggest that 25(OH)D may downregulate SREBP expression or activity via its receptor, the vitamin D receptor (VDR), thus leading to a decrease in both cholesterol and fatty acid synthesis [30,31]. Another study examined the impact of a high-fat diet on SREBP and observed that such a diet significantly elevated SREBP expression, leading to dysregulated lipid metabolism. Importantly, the study found that supplementation with 25(OH)D was effective in reversing these adverse changes [32], which side-steps the point. In contrast, androgens activate the SREBP lipid metabolic pathway and increase SCAP levels, thereby promoting the production of LDL and TC [33]. This may account for our findings in men, where androgen concentrations are higher and more active before the age of 50. In this group, the opposing effects of 25(OH)D and androgens on SREBP and SCAP likely result in stable LDL and TC levels despite rising 25(OH)D concentrations. After the age of 50, when androgen levels and activity decline, the influence of 25(OH)D on SREBP and SCAP may become more dominant, leading to a negative correlation between 25(OH)D and both LDL and TC, consistent with the overall trend.

For women, the influence of age on the relationship between 25(OH)D and LDL and TC is notably dichotomous. Up to the age of 50, there is a positive correlation between 25(OH)D and both LDL and TC. However, after this age, the correlation becomes negative. To our knowledge, this is the first report of such an age-dependent variation in the correlation between 25(OH)D and LDL and TC. One possible explanation is that high levels of active estrogen are associated with increased LDL and TC concentrations as 25(OH)D levels rise. This may be due to the fact that elevated estrogen levels increase the concentration of vitamin-D-binding protein (DBP) before menopause, which transports vitamin D and its metabolites in the bloodstream [34]. Although higher DBP levels result in an increased total vitamin D content, they may also reduce the levels of free, biologically active vitamin D [35]. Consequently, in a high-estrogen environment, the availability of active vitamin D may be diminished, even if total 25(OH)D concentrations remain relatively stable. Additionally, estrogen may facilitate the conversion of 25(OH)D to 1,25(OH)_2_D by modulating the enzyme CYP27B1, potentially leading to a reduction in 25(OH)D levels [36]. As women age and estrogen levels decline, this regulatory effect of estrogen weakens, allowing 25(OH)D to more directly influence LDL and TC levels again.

Therefore, while estrogen supplementation is recognized as a potential treatment to reduce cardiovascular disease incidence and mortality in elderly women [37], our findings suggest that it is crucial to monitor a patient’s 25(OH)D levels concurrently. This is important to prevent the simultaneous presence of elevated estrogen and 25(OH)D levels, which can synergistically raise lipid levels and potentially contribute to the development of cardiovascular diseases.

### Study Limitations

Although this study offers valuable insights into the relationship between vitamin D levels and lipid metabolism, it has several limitations that should be noted. First, the cross-sectional design of the study restricts our ability to infer causal relationships between vitamin D levels and lipid parameters. Longitudinal studies are needed to ascertain whether changes in vitamin D status have a direct effect on lipid metabolism over time. Second, the absence of sex hormone level measurements introduces potential inaccuracies in analyses related to sex and age. Additionally, the study population’s restriction to a specific geographic area (Germany) may limit the generalizability of the findings to other populations with different geographic, environmental, and genetic backgrounds. Furthermore, the study’s reliance on total vitamin D measurements, rather than free vitamin D, may not fully reflect bioactive vitamin D levels. Measurements of free vitamin D would have been of major interest. Free vitamin D represents the bioactive form of vitamin D because it can freely pass the cell membrane and activate the nuclear vitamin D receptor. Vitamin D bound to vitamin-D-binding protein, on the other hand, cannot pass the cell membrane and interact with the nuclear vitamin D receptor in most human cell types [38]. Thus, analyzing the relationship between free vitamin D and lipids would have been of major interest. Another limitation is the lack of data on other potential confounders, such as socioeconomic factors, dietary intake, physical activity, and the use of lipid-lowering medications, all of which may impact both vitamin D levels and lipid metabolism. Also, the lack of basic information (including underlying diseases, body mass index (BMI), lifestyle habits, etc.) is an issue that needs to be addressed and considered in further studies. Lastly, while the study’s large sample size is a strength, subgroup analyses by sex and age reduced the number of cases in each category, potentially affecting the statistical power of these analyses. Future research should address these limitations to enhance our understanding of the complex relationship between vitamin D and lipid metabolism; an intervention trial could also be conducted to validate the findings of this study by examining whether moderate vitamin D supplementation improves lipid profiles across different subgroups.

## 5. Conclusions

In conclusion, this cross-sectional study highlights an association between higher levels of 25(OH)D and improved lipid profiles, including lower LDL and total cholesterol, along with higher HDL levels, especially in individuals over 50 and postmenopausal women. However, due to the cross-sectional nature of the study, we cannot infer causality between vitamin D levels and lipid parameters. The absence of key clinical data, such as BMI, waist-to-hip ratio, and information on lipid-lowering medications, limits our ability to fully account for other confounding factors that may influence these associations. These findings nonetheless support the need for further research, including longitudinal and interventional studies, to better understand the role of vitamin D in lipid metabolism, particularly in relation to age- and sex-specific variations in cardiovascular risk.

## Figures and Tables

**Figure 2 nutrients-16-03936-f002:**
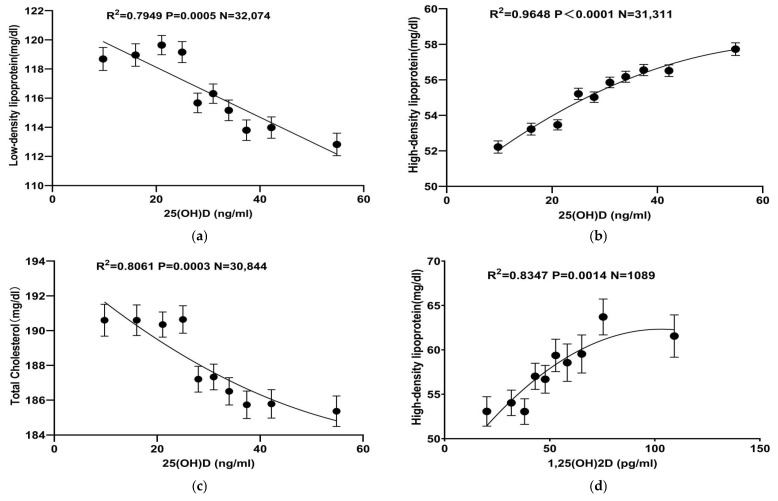
Correlation between vitamin D and lipid parameters. (**a**–**c**) depict the correlation curves between 25(OH)D and LDL, HDL, and TC. In these figures, LDL and TC concentrations decrease as 25(OH)D levels increase, whereas HDL concentration shows an opposite trend, increasing with higher 25(OH)D levels. (**d**) shows the correlation curve between 1,25(OH)_2_D and HDL. Similar to 25(OH)D, HDL showed a positive trend with 1,25(OH)_2_D. Abbreviations: 25(OH) D: 25-hydroxy-vitamin D; 1,25(OH)_2_D: 1,25-dihydroxy-vitamin D.

**Figure 3 nutrients-16-03936-f003:**
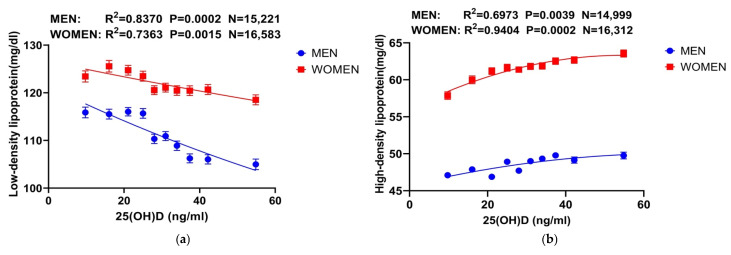
Correlation between vitamin D and lipids after grouping according to gender. (**a**–**c**) Correlation curves between 25(OH)D and LDL, HDL, and TC after gender grouping. For both males and females, the trend is consistent with the overall data, i.e., LDL as well as TC are negatively correlated with 25(OH)D, whereas HDL is positively correlated. (**d**) Correlation curve between 1,25(OH)_2_D and HDL after gender grouping. The trends were not gender-specific and were consistent with the data comprising all study participants, with HDL concentrations increasing as 1,25(OH)_2_D concentrations increased. On the other hand, vitamin D concentrations were higher in the female group than in the male group. Abbreviations: 25(OH) D: 25-hydroxy-vitamin D; 1,25(OH)_2_D: 1,25-dihydroxy-vitamin D.

**Figure 4 nutrients-16-03936-f004:**
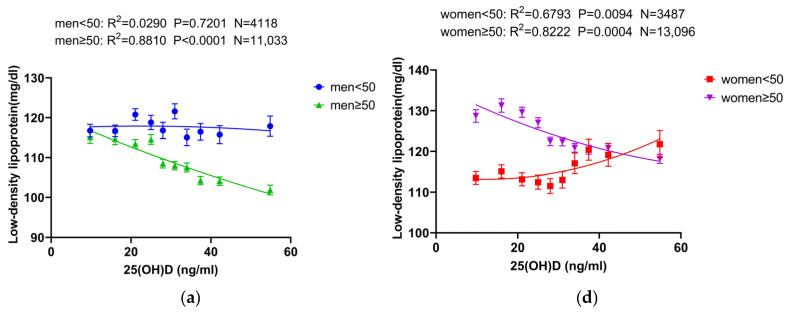
Correlation between 25(OH)D and lipids by gender and age. The correlation curves of 25(OH)D with lipids at different ages in the male group are represented in (**a**–**c**). Before the age of 50, LDL and TC were not significantly correlated with 25(OH)D, and their concentrations did not change in response to changes in the concentration of 25(OH)D; after the age of 50, LDL was negatively correlated with TC. HDL showed a positive correlation with 25(OH)D in males at any age. The correlation curves of 25(OH)D with lipids at different ages in the female group are represented in (**d**–**f**). Before 50 years of age, LDL and TC showed a positive trend of correlation with 25(OH)D, and after 50 years of age, a negative correlation was observed. HDL showed a positive correlation with 25(OH)D at any age in females. Abbreviations: 25(OH) D: 25-hydroxy-vitamin D; 1,25(OH)_2_D: 1,25-dihydroxy-vitamin D.

**Figure 5 nutrients-16-03936-f005:**
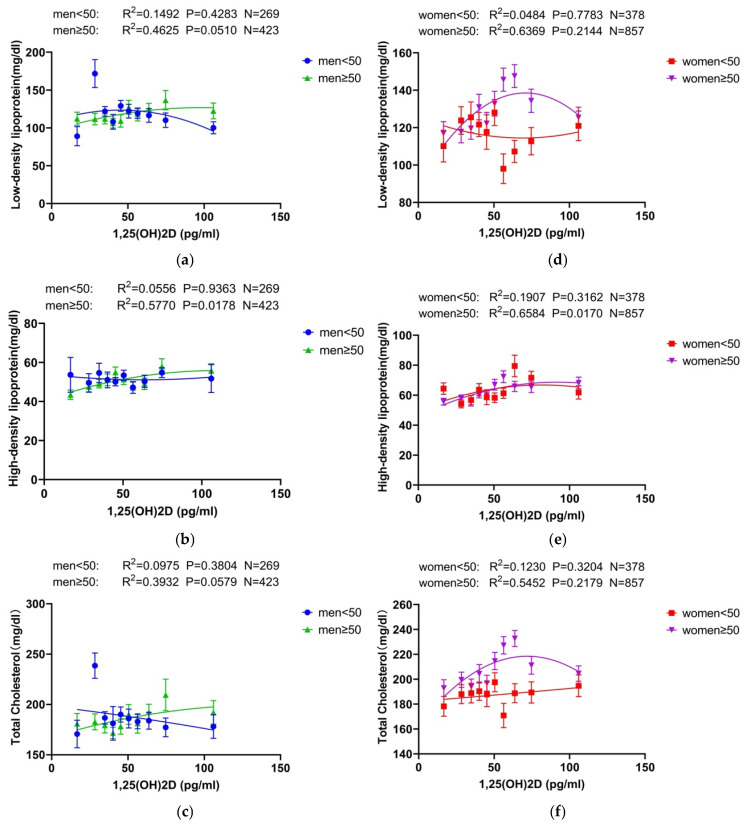
Correlation between 1,25(OH)_2_ vitamin D and lipids by gender and age. (**a**–**c**) represent the correlation curves of 1,25(OH)_2_D with lipids at different ages in the male group. (**d**–**f**) show the correlation curves of 1,25(OH)_2_D with lipids at different ages in the female group. LDL and TC were not correlated with 1,25(OH)_2_D at any time in either sex. For HDL, a correlation with 1,25(OH)_2_D was shown only after the age of 50 years in either men or women. Abbreviations: 25(OH) D: 25-hydroxy-vitamin D; 1,25(OH)_2_D: 1,25-dihydroxy-vitamin D.

**Figure 6 nutrients-16-03936-f006:**
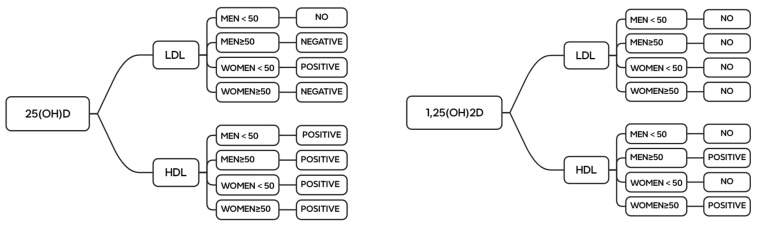
Total vitamin-D–lipid correlation after age and gender grouping. In the figure, “no” means that there is no correlation between the two, “positive” means that vitamin D is positively correlated with lipids, and “negative” means that the two are negatively correlated. Abbreviations: 25(OH)D: 25-hydroxy-vitamin D; 1,25(OH)_2_D: 1,25-dihydroxy-vitamin D; LDL: low-density lipoprotein, HDL: high-density lipoprotein.

**Table 2 nutrients-16-03936-t002:** Curve estimation of serum 25(OH)D.

Parameters	All					
	Linear		Exponential		Quadratic	
	R^2^	*p*	R^2^	*p*	R^2^	*p*
LDL (mg/dL)	0.7947	0.0005	0.7938	0.0007	0.7949	0.0003
HDL (mg/dL)	0.9225	<0.0001	0.9534	<0.0001	0.9648	<0.0001
Total Cholesterol (mg/dL)	0.7846	0.0006	0.7936	0.0004	0.8061	0.0003
**Parameters**	**MEN**					
	**Linear**		**Exponential**		**Quadratic**	
	**R^2^**	** *p* **	**R^2^**	** *p* **	**R^2^**	** *p* **
LDL (mg/dL)	0.8345	0.0003	0.8299	0.0003	0.7949	0.0002
HDL (mg/dL)	0.6681	0.0041	0.6871	0.0040	0.9648	0.0039
Total Cholesterol (mg/dL)	0.8850	<0.0001	0.8992	<0.0001	0.8061	<0.0001
**Parameters**	**WOMEN**					
	**Linear**		**Exponential**		**Quadratic**	
	**R^2^**	** *p* **	**R^2^**	** *p* **	**R^2^**	** *p* **
LDL (mg/dL)	0.7356	0.0018	0.7298	0.0024	0.7363	0.0015
HDL (mg/dL)	0.8460	0.0003	0.8782	0.0002	0.9404	0.0002
Total Cholesterol (mg/dL)	0.5564	0.0207	0.6011	0.0183	0.6411	0.0132

Grouping according to gender and description of best-fit curves of 25(OH)D versus lipid parameters (LDL/HDL/TC). Abbreviations: 25(OH)D: 25-hydroxy-vitamin D; LDL: low-density lipoprotein, HDL: high-density lipoprotein.

**Table 3 nutrients-16-03936-t003:** Curve estimation of serum 1,25(OH)_2_D.

Parameters	All					
	Linear		Exponential		Quadratic	
	R^2^	*p*	R^2^	*p*	R^2^	*p*
LDL (mg/dL)	0.1999	0.3860	0.6961	0.2223	0.7273	0.1951
HDL (mg/dL)	0.7204	0.0019	0.8034	0.0015	0.8347	0.0014
Total Cholesterol (mg/dL)	0.2644	0.1284	0.3334	0.1037	0.4734	0.0924
**Parameters**	**MEN**					
	**Linear**		**Exponential**		**Quadratic**	
	**R^2^**	** *p* **	**R^2^**	** *p* **	**R^2^**	** *p* **
LDL (mg/dL)	0.0460	0.5518	0.1478	0.4523	0.3211	0.1963
HDL (mg/dL)	0.4520	0.0332	0.4827	0.0298	0.5069	0.0166
Total Cholesterol (mg/dL)	0.1173	0.3325	0.1223	0.3026	0.1401	0.2298
**Parameters**	**WOMEN**					
	**Linear**		**Exponential**		**Quadratic**	
	**R^2^**	** *p* **	**R^2^**	** *p* **	**R^2^**	** *p* **
LDL (mg/dL)	0.1713	0.2344	0.4532	0.0672	0.6523	0.0562
HDL (mg/dL)	0.5375	0.0161	0.6011	0.0152	0.7066	0.0025
Total Cholesterol (mg/dL)	0.2376	0.1775	0.4532	0.0986	0.5526	0.0678

Grouping according to gender and description of best-fit curves of 1,25(OH)_2_D versus lipid parameters (LDL/HDL/TC). Abbreviations: 1,25(OH)_2_D: 1,25-dihydroxy-vitamin D; LDL: low-density lipoprotein, HDL: high-density lipoprotein.

**Table 4 nutrients-16-03936-t004:** Multivariate regression for serum 25(OH)D.

	All				Men				Women			
	*p*	FDR	B#	95%CI	*p*	FDR	B#	95%CI	*p*	FDR	B#	95%CI
Constant	<0.001	/	13.128	12.235~14.020	<0.001	/	12.973	11.686~14.187	<0.001	/	14.204	12.863~15.544
Gender	0.048	0.050	0.301	0.021~0.580	/	/	/	/	/	/	/	/
Age (years)	0.000	0.040	0.176	0.168~0.185	<0.001	0.013	0.175	0.164~0.186	<0.001	0.025	0.177	0.166~0.189
Seasonal grouping	<0.001	0.020	2.643	2.366~2.919	<0.001	0.025	3.207	2.825~3.589	<0.001	0.013	2.113	1.714~2.512
LDL (mg/dL)	<0.001	0.010	−0.015	−0.019~−0.012	<0.001	0.038	−0.016	−0.021~−0.011	<0.001	0.038	−0.015	−0.019~−0.010
HDL (mg/dL)	<0.001	0.030	0.058	0.050~0.066	<0.001	0.050	0.053	0.040~0.066	<0.001	0.050	0.061	0.050~0.070

The final model was adjusted for sex, age, seasonal group, LDL, and HDL. Figure 2 shows that 25(OH) vitamin D is linearly associated with LDL and HDL. Seasonal groups: The data were categorized into two groups according to the seasonal variations in vitamin D concentrations: the first group exhibited low vitamin D levels from January to April and in December, while the second group showed high vitamin D levels from May to November. Abbreviations: 1,25(OH)_2_D: 1,25-dihydroxy-vitamin D, LDL: low-density lipoprotein, HDL: high-density lipoprotein, B#: unstandardized coefficient B, FDR: false discovery rate threshold.

**Table 5 nutrients-16-03936-t005:** Multivariate regression for serum 1,25(OH)_2_D.

	All			Men			Women		
	*p*	B#	95%CI	*p*	B#	95%CI	*p*	B#	95%CI
Constant	0.258	39.319	28.900~107.538	0.369	41.179	48.789~131.147	0.540	34.074	75.056~143.204
Gender	0.937	0.179	−4.251~4.608	/	/	/	/	/	/
Age (years)	0.001	−0.193	−0.309~−0.076	0.001	−0.264	−0.422~−0.106	0.094	−0.143	−0.310~0.025
Seasonal grouping	0.189	2.759	−1.358~6.867	0.462	2.269	−3.326~7.863	0.235	3.573	−2.328~9.473
LDL (mg/dL)	0.075	−0.291	−0.611~0.029	0.267	−0.205	−0.567~0.157	0.089	−0.490	−1.055~0.075
Sqrt-LDL	0.051	6.970	−0.706~14.646	0.233	4.788	−3.088~12.665	0.059	12.352	−0.648~25.353
HDL (mg/dL)	0.149	0.742	−0.267~1.751	0.649	0.355	−1.175~1.885	0.119	1.191	−0.307~2.689
Sqrt-HDL	0.225	−9.776	−25.568~6.016	0.754	−3.627	−26.327~19.072	0.155	−17.513	−41.662~6.636

Based on our findings, Table 3 demonstrates that 1,25(OH)_2_ vitamin D is correlated exclusively with HDL. As illustrated in Figure 3, the correlation between 1,25(OH)_2_ vitamin D and HDL does not follow a typical linear pattern, independent of gender. To account for potential non-linear relationships, we incorporated the square roots of both LDL and HDL into our model. The final model was adjusted for sex, age, seasonal group, LDL, Sqrt-LDL, HDL, and Sqrt-HDL. Seasonal groups: The data were categorized into two groups according to the seasonal variations in vitamin D concentrations: the first group exhibited low vitamin D levels from January to April and in December, while the second group showed high vitamin D levels from May to November. Abbreviations: 1,25(OH)_2_D: 1,25-dihydroxy-vitamin D, LDL: low-density lipoprotein, HDL: high-density lipoprotein, Sqrt-: square root parameters; B#: unstandardized coefficient B.

## Data Availability

All data generated or analyzed during this study are included in this article. Further enquiries can be directed to the corresponding author.

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
