# Peer review of "Vitamin D Is Associated with Lipid Metabolism: A Sex- and Age-Dependent Analysis of a Large Outpatient Cohort"

_nutrients, 2024, doi:10.3390/nu16223936_

Round 1

Reviewer 1 Report

Comments and Suggestions for Authors

Major comments:

  1. Results Section: The manuscript is currently too concise, particularly in the Results section, which should be expanded to provide a comprehensive overview of the findings. This should include more detailed explanations of key trends, data interpretation, and their relevance to the research question.

  2. Statistical Analysis: Please enhance the statistical analysis by reporting the False Discovery Rate (FDR) instead of p-values, which will provide a more accurate representation of multiple comparisons and reduce the likelihood of Type I errors.

  3. Sample Origin: Provide more detailed information about the sample origins. Were all samples collected and measured in Berlin? Clarify whether the donor pool was genetically homogeneous or heterogeneous, as this is crucial for understanding the generalizability of the findings.

  4. Illustrative Figures: The inclusion of additional figures would be beneficial in conveying the main message of the manuscript. Visual representations, such as charts or graphs, would help to better illustrate key results and improve the manuscript’s overall clarity.

Minor comments:

  1. Manuscript Template: Please use the most up-to-date manuscript template from 2024, rather than the 2021 version.

  2. Tables: Incorporate the tables directly into the main body of the manuscript, rather than including them separately.

  3. Consistent Use of Abbreviations: Ensure that abbreviations are only defined once, upon first use, and applied consistently throughout the manuscript and in the tables, avoiding repeated definitions.

Reviewer 2 Report

Comments and Suggestions for Authors

The manuscript entitled “Vitamin D is Associated with Lipid Metabolism: A Sex- and Age-Dependent Analysis of a Large Outpatient Cohort” by Li et al. studies the association between Vitamin D and various lipid parameters (LDL, HDL, TC), trying to examine the reciprocal effects across different age and sex groups. The study involves a significant cohort of 47,778 participants, which provides significant statistical power.

Although authors explain in the manuscript the relationship between 25(OH)D and lipids, particularly LDL and HDL, with a focus on the sex- and age-dependent differences, there are still significant areas of improvement that need to be addressed and the study, as also pointed out by the authors in a specific section, has significant limitations that could alter the results and their strengths. 

Major areas:

-       Authors mention that the study population was taken from the “general healthy population”. What does this mean? It would be helpful to add more details about the baseline health status of participants. Did they have known conditions (e.g., diabetes, hypertension)? This could impact the generalizability of your results. A specific table or a revised table 1 with these information could really help the overall quality of the study. 

-       Use of covariates: while LDL, HDL, age, and seasonal variation are included in the regression models, authors should better clarify why other factors such as BMI, physical activity, and smoking weren't included, or if they were considered and later excluded. If this was not evaluated, authors should expand on the results including these factors (especially smoking) in the revised statistics.

-       In the results section, some of the results are really hard to read and follow. The age- and sex-dependent results are a bit complex. Authors need to revise language and the narrative flow by first discussing the overall trends and then diving into the subgroup findings. For example:

"Overall, 25(OH)D was inversely associated with LDL and TC but positively associated with HDL. However, when stratified by sex and age, these relationships showed notable variation." Then expand on these variations separately for men and women across age brackets.

-       Authors have correctly pointed out that the study has significant limitations, but I would suggest one more, which are unmeasured confounders (such as socioeconomic factors, diet, and lifestyle habits). The limitations section then needs to be revised. 

-       Do authors have data on waist circumference as well as BMI?

-       Given the major limitations to the study, conclusions need to be toned down. Authors cannot state that “These findings highlight the broader role of 25(OH)D in regulating lipid metabolism” as well as that “These results suggest that optimizing 25(OH)D levels could be a valuable strategy in managing dyslipidemia, particularly in older adults and postmenopausal women” unless the study is really improved and some of the study limitations are actually dealt with. Indeed, authors should know that how vitamin D optimization might be integrated into lipid-lowering strategies or cardiovascular risk management, particularly for older adults and postmenopausal women, is not well demonstrated in the current literature.

Minor areas:

-       Regarding the abstract, the first few sentences briefly touch on vitamin D's role in calcium homeostasis and cardiovascular health. It would be more engaging to directly state the research question or objective early in the abstract to immediately convey the study’s purpose.

-       Also, regarding the abstract, the methods section is very brief and should be revised. 

-       Overall, the entire structure of the abstract needs to be revised, also considering the comment regarding conclusions of the study. 

-       I would suggest that authors consider as reference https://doi.org/10.3390/nu14020347 , a previous study that underlined significant differences in females regarding Vitamin D and Monocyte-to-HDL-ratio, an inflammatory marker that could be useful for the introduction and/or discussion. 

-       While authors mention in the introduction previous studies with conflicting results on vitamin D and lipid metabolism, expanding on specific gaps would strengthen the rationale of the study. What were the most significant differences and how do they link to the present study? Smaller sample sizes? Lack of age/sex differentiation? Etc.

-       Some of the correlations (especially the age and sex-specific ones) might benefit from additional subgroup figures. For instance, graphs that show the difference in vitamin D-lipid correlations before and after 50 years of age for each sex separately could help the reader quickly grasp the key points.

-       In the discussion, the molecular pathways (e.g., SREBP and SCAP) are mentioned briefly. Authors could expand on how previous studies have shown vitamin D modulating these pathways, referencing relevant research to deepen the discussion.

-       Authors have noted the need for longitudinal studies, which is excellent. Perhaps, they could also mention the need for intervention studies that assess whether vitamin D supplementation improves lipid profiles in different subgroups (e.g., postmenopausal women or older men).

Comments on the Quality of English Language

Overall quality of English is acceptable, but must be improved to help readers navigate the results and discussion.

Reviewer 3 Report

Comments and Suggestions for Authors   In this work the authors analyzed the link between vitamin D and lipid metabolism. The human sample appears to be fairly well distributed between the sexes, the age is wide (59.00±19.51), no particular pathologies or habits are reported (e.g. smoking, physical exercise, or other). A more accurate analysis of the sample could give further ideas to the authors, in particular for physical exercise, an aspect that the authors have taken into account in their work. Since the authors emphasize the aspect of menopause, they could dose estrogens and verify a relationship between vitamin and hormone levels  

Round 2

Reviewer 1 Report

Comments and Suggestions for Authors

1. Calculating FDR values does not harm, please add this info.

2. Please fully adapt the style of the manuscript template.

Reviewer 2 Report

Comments and Suggestions for Authors

Authors addressed all raised concerns positevely.

The manuscript has been significantly improved.

Author Response

Thank you for your guidance and affirmation!